# High-Efficiency Simplified Orange and White Organic Light-Emitting Devices Based on a Platinum(II) Complex

**DOI:** 10.3390/mi14010065

**Published:** 2022-12-27

**Authors:** Qing Zhao, Dongfang Zhao, Xinchen Zhang

**Affiliations:** 1Central China Normal University Wollongong Joint Institute, Central China Normal University, Wuhan 430079, China; 2College of Physical Science and Technology, Central China Normal University, Wuhan 430079, China

**Keywords:** high-efficiency, simplified structure, organic light-emitting diodes, white light emission

## Abstract

We demonstrated efficient simplified orange and white organic light-emitting devices based on a platinum(II) complex Tetra-Pt-N. The maximum current efficiency achieved from the optimized orange device was 57.6 cd/A. The emission mechanism for the system of Tetra-Pt-N doped into 4,4’-bis(arbazole-9-yl)biphenyl was discussed. Moreover, a high-efficiency and simplified white device was fabricated by introducing an ultra-thin blue phosphorescent emission layer. The white device with a maximum current efficiency of 41.9 cd/A showed excellent stable spectra and low efficiency roll-off.

## 1. Introduction

Organic light-emitting diodes (OLEDs) have a great potential to become the next-generation flat-panel displays and solid-state lightings thanks to their low power consumption, high performance and flexible display in larger areas [1,2,3]. However, achieving high efficiency in OLEDs is still an open research problem, which prevents universal production in long-term development [4,5]. In order to improve the efficiency, extraordinary efforts have been made using a variety of novel organic materials or by adopting novel device structures [6,7].

The transition metal ions such as Ir(III) and Pt(II) effectively induce spin–orbit coupling and make singlet–triplet excited states sufficiently mixed. The phosphorescent materials can achieve 100% internal quantum efficiency due to the ability of capturing both the single and triplet excitons [8]. The performances of OLEDs using Ir(III) complexes have been greatly improved over the last decade [9,10,11,12]. Wang et al. demonstrated the performance of green phosphorescent OLEDs based on bis(2-phenylpyridine) (acetylacetonate) iridium(III) with an efficiency of 78.0 lm/W at 100 cd/m^2^ [9]. Sasabe et al. reported on highly efficient green OLEDs based on fac-tris(2-phenylpyridine) iridium(III) with a current efficiency of 84 cd/A [10]. Although platinum(II) complexes are considered as other promising efficient phosphorescent emitters due to highly emissive excited states, the performances of most reported Pt(II) OLEDs in the literature were still significantly inferior to the best iridium(III) OLEDs [13,14,15,16,17]. Moreover, most of the reported works placed great emphasis on material synthesis rather than on the design of the device structure and the exploration of emission mechanisms [18,19,20]. Therefore, it is of enormous significance to investigate the structure and the device physics of OLEDs using a platinum(II) complex.

In this work, we demonstrated simplified bilayer orange and white OLEDs based on an orange-emitting platinum(II) complex Tetra-Pt-N. By optimizing the doping concentration and the doped emitting layer thickness, the excited-state charge transfer and exciton dissociation were reduced and performances of devices were improved. The orange and white OLEDs based on a simplified bilayer structure have a maximum current efficiency of 57.6 cd/A and 41.9 cd/A, respectively.

## 2. Experiment

Clean glass coated with indium tin oxide (ITO) was used as the transparent anode. A quantity of 2 nm MoO_3_ was deposited on ITO to improve the hole injection. A common host material 4,4′-bis(carbazol-9-yl)biphenyl (CBP) with high carrier mobility served as both the hole-transporting layer (HTL) and the host for the orange phosphorescent material. Tetra-Pt-N doped into CBP acted as the orange phosphorescent emission layer. 2,2′,2′′-(1,3,5-benzinetriyl)-tris(1-phenyl-1- H-benzimi-dazole) (TPBi) functioned as a hole-blocking layer and an electron-transporting layer. Finally, 0.8 nm LiF covered by 100 nm Al was used as the cathode. The organic materials were purchased from Luminescence Technology Corp. All organic layers were grown in succession using high vacuum (2 × 10^−4^ Pa) thermal evaporation at a rate of 0.08–0.15 nm/s. The layer thickness and the deposition rate of materials were monitored in situ using an oscillating quartz thickness monitor. The electroluminescence (EL) spectra and Commission International de L’Eclairage (CIE) coordinates of the OELDs were measured using a PR655 spectroscan spectrometer. The current–voltage and luminance–voltage characteristics were measured simultaneously with the measurement of the EL spectra by combining the spectrometer with a programmable Keithley 2400 voltage–current source. All measurements were carried out at room temperature under ambient conditions.

## 3. Results

The structure (a) and the energy level diagram (b) of the simplified bilayer orange devices are shown in Figure 1. In order to determine the optimized concentration of orange phosphorescent dye, we fabricated four devices with the following structure of ITO/MoO_3_ (2 nm)/CBP (60 nm)/CBP: Tetra-Pt-N (X wt%, 10 nm)/TPBi (30 nm)/LiF (0.8 nm)/Al. Here, X = 1, 5, 10 and 20 corresponding to the devices A-D, respectively. In this configuration, the excitons are well confined in the emission layer, which contributes to the wide energy gap materials of CBP and TPBi with high triplet energies.

Figure 2 shows the EL performances of the four devices, and the inset in Figure 2b shows the normalized EL spectra of the device A at different luminance. The maximum current efficiency increases from 36.4 cd/A (28.6 lm/W) to 43.1 cd/A (33.8 lm/W) with the doping concentration of Tetra-Pt-N increasing from 1 to 5 wt%, and then decreasing to 37.3 cd/A (29.3 lm/W) with the doping concentration increasing to 20 wt%. The maximum current efficiency of 43.1 cd/A and power efficiency of 33.8 lm/W are achieved for the device with a doping concentration of 5 wt%. Generally speaking, the doping concentration of Tetra-Pt-N has little influence on device performance.

Figure 2b shows the normalized EL spectra at the voltage of 9 V of the four devices. The spectra of the devices show a main peak at around 548 nm originating from Tetra-Pt-N. The redshift in the EL spectra with increasing doping concentrations has always been observed in many works. According to the results of S.R. Forrest et al. [21], the peak emission wavelength shift is found to be due to strong polarization effects. The normalized EL spectra of the devices E–H at 9 V are shown in the inset of Figure 2a. A weak emission peak from the CBP host at 400 nm can be observed in device A with the lower doped concentration of 1 wt%, and the intensity increases with applied voltage. The observation is attributed to the emission sites easily reaching saturation and inefficient energy transfer from CBP to Tetra-Pt-N. With the doped concentration of Tetra-Pt-N increasing, the emission of CBP disappears. In addition, the spectra slightly show redshift with an increase in doping concentration.

Performances of this kind of simplified bilayer device can be further improved by fixing the optimized concentration at 5% and adjusting the thickness of the orange emission layer (EML). The structures of simplified bilayer devices are as follows: ITO/MoO_3_ (2 nm)/CBP [(70-Y) nm]/CBP: Tetra-Pt-N (5 wt%, Y nm)/TPBi (30 nm)/LiF (0.8 nm)/Al (100 nm). Here, Y was chosen as 3, 8, 20 and 70, which corresponds to devices E–H, respectively. Figure 3 shows current density–voltage–current efficiency (a), current efficiency–luminance (b), and characteristics of the devices E–H. It can be seen that the current density markedly decreases when Y increases from 2 nm to 70 nm. A decrease in current density is mainly because Tetra-Pt-N acts as a deep trap for holes which also can be confirmed by the true fact that the highest occupied molecular orbital (HOMO) level of Tetra-Pt-N is 1.5 eV lower than that of CBP. There are two possible mechanisms for electrically exciting phosphorescence: energy transfer from CBP and charge trapping on dyes. For the system of Tetra-Pt-N doped in CBP, energy transfer is the dominant emitting mechanism. The inset in Figure 3a shows the normalized EL spectra of the devices E-G at the voltage of 9 V. Note that the peaks of the spectra show redshift from 540 nm to 555 nm with Y increasing. The redshift that occurred in the spectra is attributed to the increasing of the space–charge density in the EML because more holes are trapped for thicker EMLs [22]. Moreover, the microcavity effect causes the spectra to shift to the red when Y increases.

Performances of the devices E–F are summarized in Table 1. As can be seen, when the thickness of orange EML increases from 3 nm to 70 nm, the maximum luminance increases from 24,690 cd/m^2^ to 96,870 cd/m^2^, gradually. The maximum efficiency increases from 35.3 cd/A (27.7 lm/W) to 57.6 cd/A (45.2 lm/W) as Y increases from 3 nm to 8 nm, then decreases to 44.3 cd/A for device G. The high efficiency of device F is attributed to the efficient hole-trapping effect of Tetra-Pt-N which improves the electron–hole balance in the emission layer. From the energy level diagrams of the device, we can see that the main excitons’ recombination zone is located at the interface of EML and TPBi. The following processes can occur for the excitons formed at the narrow zone near the interface. The majority of excitons are transferred to the Tetra-Pt-N-emitting orange light. For the system of Tetra-Pt-N doped in CBP, both the HOMO and the lowest unoccupied molecular orbital (LUMO) levels are offset, so excited-state charge transfer may occur resulting in exciton dissociation [23]. While EML becomes thicker, the phosphorescence quenching is more serious which leads to a lower efficiency for devices G and H. Though device H shows a slight efficiency roll-off due to the broader carrier recombination zone and relative charge-balancing, the turn-on voltage is about 1.5 V higher than the other three devices. For device H, the majority of the holes reside primarily on Tetra-Pt-N, far from the EML/TPBi interface where the electrons accumulate at the lower voltage. The hole-trapping effect of Tetra-Pt-N in device H leads to a higher turn-on voltage.

White light emission can be obtained by doping multiple dyes into one single EML or by using multiple EMLs [24,25,26,27,28]. However, doping is a sophisticated device fabrication process where it is difficult to precisely control the relative concentration of guest/host doping, especially for white OLEDs. To simplify the device structure, an efficient simplified white device was fabricated by introducing an ultra-thin blue phosphorescent EML. Encouraged by the significantly enhanced EL performance in the orange OLEDs, we successfully fabricated a simplified white device by incorporating an ultra-thin blue EML with an optimal emission thickness. The structure of the white device is ITO/MoO_3_ (2 nm)/CBP (62 nm)/CBP: Tetra-Pt-N (5 wt%, 8 nm)/FIrPic (0.2 nm)/TPBi (30 nm)/LiF (0.8 nm)/Al (100 nm). Figure 4 shows the EL performances of the white device. The maximum current efficiency of the white device is 41.9 cd/A, and retains 38.8 cd/A at the luminance of 2000 cd/m^2^ as well as 35.4 cd/A at 5000 cd/m^2^, respectively. The low efficiency roll-off can be attributed to the ambipolar transport of the host material, leading to balanced charge injection and reduced triplet–triplet annihilation. Consequently, the white device achieves a high power efficiency of 35.0 lm/W. 

The normalized EL spectra of the white device at different luminance are shown in Figure 4d. As can be seen, the white device has very stable spectra with only a slight shift in color coordinates. The CIE coordinates change from (0.337, 0.459) at 4000 cd/m^2^ to (0.345, 0.470) at 18,000 cd/m^2^. The reason why the spectra are extremely stable is attributed to the balanced charge carrier injection. As we can see, the intensity of the orange emission increases relative to that of the blue emission from FIrPic, with luminance increasing. The shift in spectra is mainly attributed to the change in the exciton recombination zone due to higher electron mobility at higher applied voltages [29]. The operational emission mechanism of the white device is shown in Figure 5. From the energy level diagrams of the device, we can see that the main exciton recombination zone is located at the interface of the orange emission layer and TPBi. Energy transfer from FIrPic to Tetra-Pt-N could also occur because the triplet energy level of FIrPic is much higher than that of Tetra-Pt-N. At a higher applied voltage, more electrons are injected into the orange emission layer with increasing luminance, leading to the orange emission intensity gradually increasing. Additionally, the emitting sites of FIrPic become saturated at higher luminance, causing the blue emission intensity to relatively decrease.

## 4. Conclusions

In summary, we have successfully investigated efficient simplified bilayer orange and white OLEDs based on an orange-emitting platinum complex Tetra-Pt-N. Wide energy gap materials of CBP and TPBi with high triplet energies were used as HTL/host and ETL, respectively. A maximum current efficiency of 57.6 cd/A (45.2 lm/W) was obtained by adjusting the doping concentration and the thickness of the orange emission layer. The white device with a simplified bilayer structure has excellent stable spectra and low-efficiency roll-off.

## Figures and Tables

**Figure 1 micromachines-14-00065-f001:**
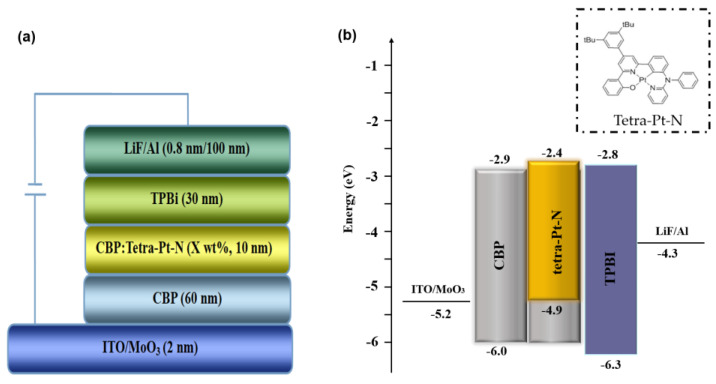
The structure (**a**) and energy level diagrams (**b**) of the devices A-D. The inset shows the molecular structure of Tetra-Pt-N.

**Figure 2 micromachines-14-00065-f002:**
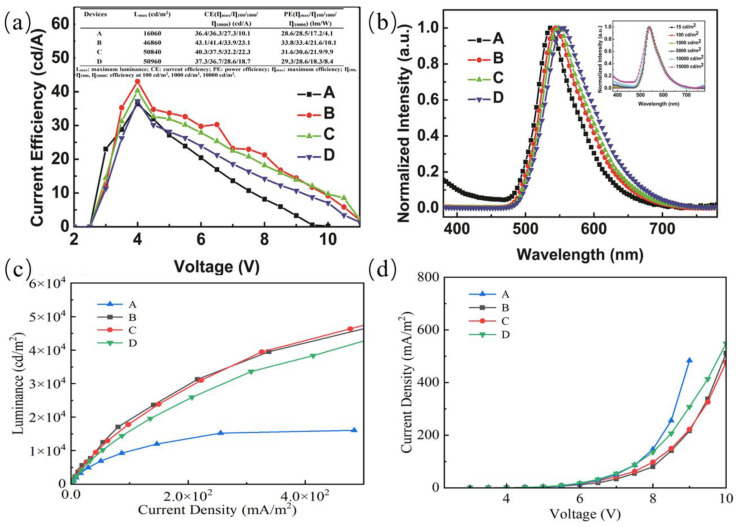
Current efficiency–voltage (**a**), normalized EL spectra at the voltage of 9 V (**b**), luminance–current density (**c**) and current density–voltage (**d**) characteristics of the devices A–D. The inset in (**b**) shows the normalized EL spectra of the device A at different luminance.

**Figure 3 micromachines-14-00065-f003:**
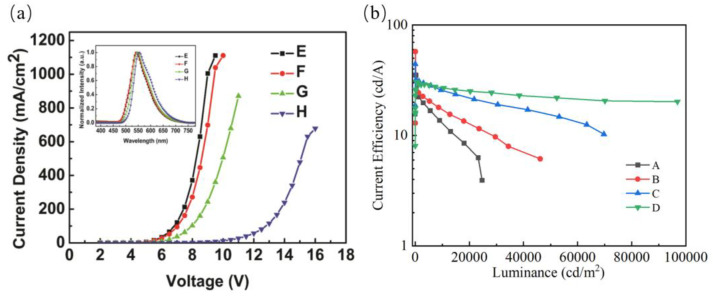
Current density–voltage (**a**) and current efficiency–luminance (**b**) characteristics of the devices E–H. The inset of Figure 3a shows the normalized EL spectra of the devices E–H at 9 V.

**Figure 4 micromachines-14-00065-f004:**
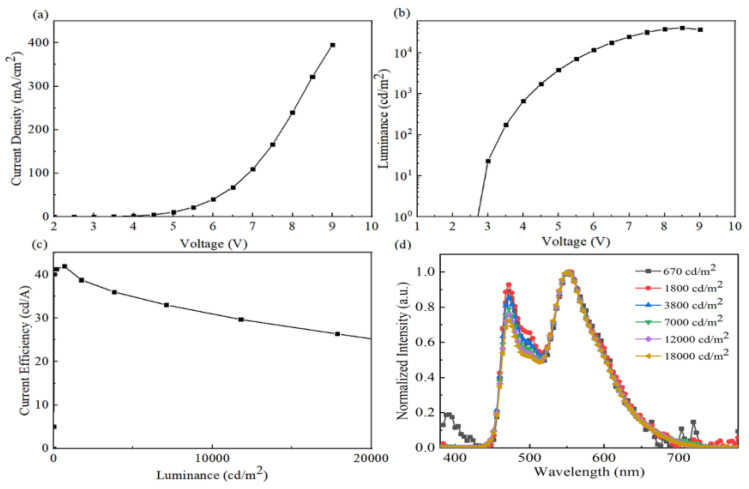
Current density–voltage (**a**), luminance–voltage (**b**), current efficiency–luminance (**c**) and normalized EL spectra at different luminance (**d**) characteristics of the white devices.

**Figure 5 micromachines-14-00065-f005:**
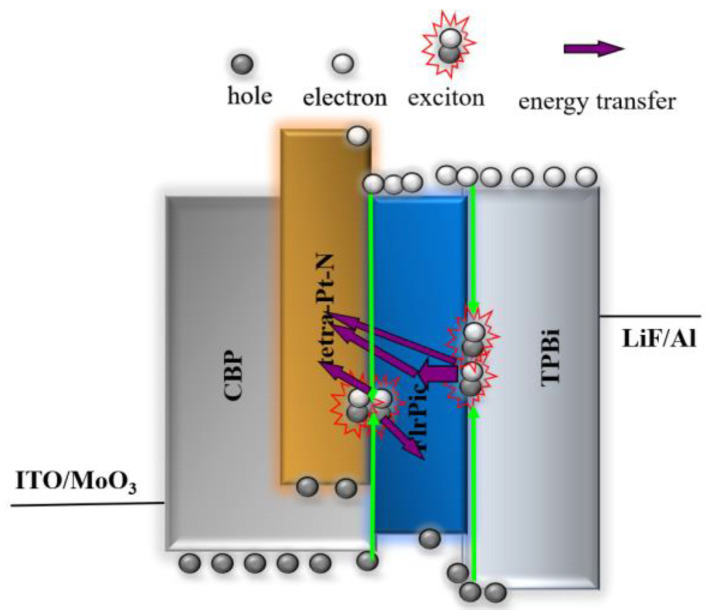
The emission mechanisms for a white OLED.

**Table 1 micromachines-14-00065-t001:** The performances of devices E–F.

Devices	Turn-on Voltage (V)	L_max_ (cd/m^2^)	CE(Ƞ_max_/Ƞ_100_/Ƞ_1000_/Ƞ_10000_) (cd/A)	PE(Ƞ_max_/Ƞ_100_/Ƞ_1000_/Ƞ_10000_) (lmW)
Device E	3.51	24,690	35.3/35.1/23.6/13.7	27.7/27.1/17.3/5.8
Device F	3.41	46,150	57.6/54.7/24.6/17.1	45.2/42.4/15.9/8.2
Device G	3.45	69,690	44.3/36.7/31.2/25.7	34.8/27.2/18.7/11.5
Device H	4.95	96,870	30.6/26.0/30.1/26.9	11.9/11.7/10.9/7.4

L_max_: maximum luminance; CE: current efficiency; PE: power efficiency; Ƞ_max_: maximum efficiency; Ƞ_100_, Ƞ_1000_, Ƞ_10000_: efficiency at 100 cd/m^2^, 1000 cd/m^2^, 10,000 cd/m^2^.

## Data Availability

Not applicable.

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
