# Peer review of "High-Efficiency Simplified Orange and White Organic Light-Emitting Devices Based on a Platinum(II) Complex"

_micromachines, 2022, doi:10.3390/mi14010065_

Round 1

Reviewer 1 Report

The authors reported yellow and white OLEDs with simple device structures. Good device performances were achieved. However, further improvement in result presentation is still needed before the manuscript can be accepted to be published in Micromechanics.

1. In the inset of Figure 2, the authors should explain the terms in detail, such as [cd A-1/%].

2. More characteristics of Devices A to D, such as the luminance-current density-voltage curves, should be presented.

3. Similarly, efficiency-luminance characteristics of Devices E to H should be shown.

4. In Figure 5d, spectra at lower luminance, such as those lower than 1000 cd m-2, should be shown.

5. The red shift in EL spectra with concentration shown in Figure 3 should be explained.

6. Lines 112-114, the explanation for the redshift in EL spectra with EML thickness should be more detailed.

7. Molecular structure of Tetra-Pt-N should be shown.

8. The style of units should be consistent throughout the manuscript, such as cd/A and cd A-1.

9. Some errors should be removed, such as in line 77.

Reviewer 2 Report

Review Comments

The document is well structured, results are well presented. however, major comments are to be taken into account before I recommend the publication of the manuscript.

1-The title could be better: example

High-efficient simplified orange and white organic light-emitting devices
based on a platinum(II) complex Tetra-Pt-N

2-line 23-24

‘’But the high efficiency and
the complicated device structure are still problems’’

You can write for example: achieving high efficiency in OLEd is still an open research problem …and cite references here. 

-3-better to continue the previous section, by citing two pertient articles like for example: ‘’Spirobifluorene-based hole-blocking material with enhanced efficiency through hole and exciton confinement in blue fluorescent OLEDs‏

J Lee, SC Kim, JY Lee - Dyes and Pigments, 2022’’ and another one.

You also need to mention exactly what challenges in the fabrication are faced nowadays.

4-in figure 2, the spectra
show redshift with the increasing of the doping-concentration,

What is the physical explanation to this?

5- The spectra are slightly different in figure 3, can the author give more details, like for example which dopping gives better peaks?

Statitically, can you calculate the difference between the three curves:for example: 15000cd/m3 red curve and 10000cd/m3 green curve, these curves have a difference of 5000, what is the difference in all the peaks obtained?

Than try to imnterpret why you obtained this ratio?

6-the maximum efficiency increases from 39.5
cd/A (27.6 lm/W) to 59.1 cd/A (46.4 lm/W) as Y is increasing from 3 nm to 8 nm, then the authors mention that there is an increases to 44.3 cd/A for device G. The argument that the efficiency increases while y increases is not clear?

How can this be explained, with respect to the penetration depth? What happens after 8nm? The saturation?

7-are there any current contribution due to dissociations of excitons?

is the increase of the efficiency due to some current from the excitons dissociations?

8- what about the carrier lifetime in these devices? Are they different device E, F…etc? is the efficiency related to the carrier lifetime?

9- in couple of occasions, the authors mentionned about the triplet triplet annihilation, the exciton dissociation. What about the possible recombination effect between the carries, is the efficiency related to the low recombination effect in some devices fabricated here?

experimentally, can this be controlled? (the recombination e-holes)

10-finally, a small theoretical paragraph as explanation  to these experimental results from the litterature, would be a plus : see for example

Volterra series analysis of the photocurrent in an Al/6T/ITO photovoltaic device

N Boutabba, L Hassine, N Loussaief, F Kouki, H Bouchriha

Organic Electronics 4 (1), 1-8

Round 2

Reviewer 2 Report

The author addressed my comments

I recommend its publication